# Comparison of Cortisol Concentrations in Different Matrices in Alpine Ibex (*Capra ibex*) at the Zoo

**DOI:** 10.3390/ani13152491

**Published:** 2023-08-02

**Authors:** Marjan Kastelic, Gordana Gregurić Gračner, Iztok Tomažič, Pavel Kvapil, Mojca Harej, Alenka Dovč

**Affiliations:** 1Zoo Ljubljana, Večna Pot 70, 1000 Ljubljana, Slovenia; marjan.kastelic@zoo.si (M.K.); pavel.kvapil@zoo.si (P.K.); mojca.harej@zoo.si (M.H.); 2Institute for Poultry, Birds, Small Mammals and Reptiles, Veterinary Faculty, University of Ljubljana, Gerbičeva 60, 1000 Ljubljana, Slovenia; 3Department of Animal Hygiene, Behaviour and Animal Welfare, Faculty of Veterinary Medicine, University of Zagreb, Heinzelova 55, 10000 Zagreb, Croatia; ggracner@gmail.com; 4The Group for Biological Education, Department of Biology, Biotechnical Faculty, University of Ljubljana, Večna Pot 111, 1000 Ljubljana, Slovenia; iztok.tomazic@bf.uni-lj.si

**Keywords:** animal welfare, zoo animals, Alpine ibex (*Capra ibex*), sampling, cortisol

## Abstract

**Simple Summary:**

Zoos help to rescue, rehabilitate, and care for wildlife. Approximately fifteen years ago, the Alpine ibex (*Capra ibex*) was successfully reintroduced into the Alps. There are at least 17 zoos throughout Europe that keep Alpine ibexes in captivity. To ensure success, the staff must understand the animal’s specific requirements and provide leading managers and researchers to investigate strategies such as feed quality control, successful reproduction, and appropriate social group structure. The animal welfare strategy is an important factor for success. Appropriate and stress-free animal capture and sampling methods are especially important. Non-invasive sampling methods are increasingly being used in veterinary practice. Cortisol (C) concentrations can be measured in various matrices, such as saliva, hair, and faeces. Blood collection using venipuncture, which is stressful, can be replaced by collection with kissing bugs (Triatomines) or medicinal leeches (*Hirudo medicinalis*). In this article, C concentrations in different matrices were compared in Alpine ibexes at the zoo. Differences according to the sex, age, and season of sampling were also compared.

**Abstract:**

The usefulness of blood collection using venipuncture versus kissing bugs or medicinal leeches and the collection of saliva, faeces, hair, urine, and tears for measuring “immunoreactive” C (iC) concentration in Alpine ibexes was verified using commercial enzyme immunoassays. The mean value of serum C was highest in serum collected using venipuncture and lowest in serums collected using kissing bugs. Statistically significant differences were observed between venipuncture and kissing bugs and between leeches and kissing bugs. However, no statistically significant difference was found in C concentrations between samples collected with venipuncture and those collected with leeches. The highest mean value of C concentration was measured in serum (all three methods), followed by that in hair and faeces, and the lowest mean value was found in saliva. Statistically significant differences were found between saliva and faeces samples and between saliva and hair samples. The difference between the concentrations for faeces and hair was not statistically significant. A significant difference in C concentration between males and females was found in saliva. A significant difference in C concentration among different ages was measured in serum obtained using venipuncture in all three groups and in faeces between the groups older than ten years and younger than 10 months. Highly significant differences in C concentrations were also found between hair sampled in summer and hair sampled in autumn. Collecting tear and urine samples is a laborious procedure and is therefore less acceptable for C determination. Due to the small number of samples, statistical values are not given for these two matrices.

## 1. Introduction

The reintroduction of species whose natural populations are declining into their original range is key to the survival of endangered species [1]. This is true for Alpine ibexes (*Capra ibex*). They used to inhabit the Alpine region in Central Europe, but were reduced to Gran Paradiso National Park in the northwestern Italian Alps. Measures to protect ibex populations were introduced in the nineteenth century, but neither antipoaching laws nor captive breeding and translocations halted the decline of the ibex population. The original article by Girtanner dates from 1878 [2]. They were successfully reintroduced at the beginning of the 20th century in the Alps of Austria, France, Germany, Italy, Slovenia, and Switzerland at altitudes between 1600 and 3200 m [3]. The re-establishment of the Alpine ibex across the Alps represents one of the most successful species reintroductions. The Alpine ibex recovered from about 100 individuals to about 50,000 individuals across the Alps in approximately one hundred years [1]. The successful reintroduction of this population can serve as a case model [2,4].

Accredited zoos frequently work to help rescue, rehabilitate, and care for wild animals [5]. Throughout Europe, there are at least 17 zoos that keep Alpine ibexes in captivity [6]. Animals in zoos are susceptible to undue stress situations and various metabolic, parasitic, or infectious diseases. The reason for this could be the greater concentration of animals, frequent contact with humans, and potentially inadequate nutrition at certain life stages. Strategies related to the health status, control of feed quality, successful reproduction, adequate social group structure, and welfare of individual animals in captivity must be ensured [7,8]. The introduction of non-invasive sampling procedures, whether to diagnose disease [9] or for the measurement of stress hormone levels [10], is an important animal welfare strategy, and should be integrated into zoo management.

Non-invasive procedures are on the rise in veterinary practice [11], and include infrared thermography [12], alternative methods of blood sampling using kissing bugs [13,14] and leeches [15], and sampling from various matrices other than serum [16]. In mammals, minimally invasive sampling includes matrices such as faeces, hair, saliva, and urine [17,18,19,20,21,22]. Milk [20,23,24] and tear [25] collections have also been performed.

Blood sampling is still the standard method for diagnostic purposes. The concentration of C is much higher in blood than in other matrices [26,27]. However, blood sampling is often a difficult and stressful method, especially when wild and zoo animals are involved. As a consequence of the stress caused to animals during capture, possible differences in C concentration can be measured [28]. For these purposes, researchers have developed a unique blood collection technique. Kissing bugs (Triatomines: *Rhodnius prolixus*, *Triatoma infestans,* and *Dipetalogaster maxima*) are commonly used as “live syringes”, primarily to avoid stress. The use of kissing bugs offers great advantages over traditional veterinary blood collection. The puncture caused by the mouthparts of kissing bugs is much smaller than that caused by a needle. The most important advantages are the possibility of blood collection without anaesthesia or stress and the ability to collect blood from animals whose veins are inaccessible [13]. Likewise, blood collected with leeches can be used for quantitative diagnostic procedures such as haematological and biochemical analyses [15] or detection of antibodies against different infectious diseases, such as, for example, tick-borne encephalitis virus [29].

Repeat blood sampling is not feasible in most non-domesticated species, so non-invasive urine and faecal metabolite studies are performed. Although urine and faecal samples can be collected to assess for, e.g., reproductive status in captive animals, difficulties in urine collection limit their use for testing in free-ranging animals. Therefore, faeces is the most practical choice of sample [30].

Hair glucocorticoids are increasingly popular biomarkers used as measures of stress in numerous research areas and in various animal species. Unlike other matrices, the analysis of C in hair is not affected by circadian fluctuations in the hormone or by factors that cause short-term fluctuations. By shaving and recollecting hair samples from the same site, C can be tracked over weeks or months, as hair samples provide a “window into the past”. This allows retrospective study of C production [18]. Recent findings [31] refute the hypothesis that cortisol is stored in growing hair and remains there for a long period of time. The authors demonstrated that glucocorticoids in hair are not static but rather diffuse in and out. They noted that traces of C remaining in the hair cannot be used as a marker of past stress.

Salivary cortisol levels indicate the current glucocorticoid status and are subject to circadian fluctuations in a similar manner to blood cortisol levels. Concentrations in saliva are approximately 10-fold lower than those in plasma. As samples can be collected non-invasively, salivary cortisol concentrations can be used to assess short-term stress [32,33].

Cortisol in blood is mainly bound to binding proteins; only a small part is unbound C. This “free” C can pass through membranes, and the C in saliva is a parameter of the free C fraction (the receptor-active fraction) in blood. For this reason, measuring salivary C can help monitor its unbound free concentrations in serum [34,35]. Protein-bound C is too large and too polar to cross cell membranes and cannot reach intracellular receptors. It accounts for more than 95% of total serum C levels [35].

However, there is a lack of studies on the presence and significance of C in tears. Studies in horses [25], sheep [16] and seals have been described [36]. Tear production in the Alpine ibex can be determined using the Schirmer tear test [37]. In horses, tears were collected by invasively placing plain glass capillary tubes in the ventral cul-de-sac of the conjunctiva overlying the third eyelid without topical anaesthesia [25]. Tear collection by capillary tubes has also been described for sheep [16]. However, there is no literature on the collection of tears in ibexes.

Some physiological measurements are difficult to obtain safely in untrained zoo animals without anaesthesia, and the procedure is also very disruptive. Training zoo animals for sampling is performed in some zoos, and may increase the chances of physiological data being able to be obtained for husbandry and research purposes [38]. The science of zoo animal welfare follows the principle that animal control is essential for good welfare and that control over the environment can be achieved by providing animals with choices in the environment [39].

In this article, we compared C concentrations in different methods of blood collection, saliva, faeces, hair, urine, and tears from the Alpine ibex in zoos. In the ibex group, we also compared differences in sex, gender, and season of sampling. The aim of our study was to find non-invasive sampling methods for ibexes, allowing the determination of stress in the group by measuring C or glucocorticoid metabolites (FGCMs) in faeces. We chose ibexes because little is known about C concentrations and their metabolites in this species, although they are quite common in zoos. Indirectly, the study could improve the welfare of ibexes in zoos.

## 2. Materials and Methods

Thirteen male and sixteen female Alpine ibexes (*Capra ibex*) of different ages from the Ljubljana Zoo were included in this study. All ibexes were kept under similar housing conditions and fed with feed from the same manufacturer. The health status of this group was monitored regularly, with samples being taken for C, and routine examinations being performed for other diagnostic tests.

### 2.1. Collection of Samples

Each ibex was captured and placed in a separate place from the other animals in the group and then manually restrained. Six different samples were taken from each of them in the course of routine clinical procedures. Saliva and hair were taken first. The whole procedure took at most one minute. Venipuncture, blood collection with kissing bugs, and leech collection were performed at the same time by different people. Two people performed venipuncture, two people collected blood samples with kissing bugs, two people collected blood samples with leeches, and two people collected tears. Blood samples with a volume of 2.0 mL were taken using G18 needles using venipuncture of the vena jugularis (V) and collected in microtainer tubes using a serum separator (Becton Dickinson, Heidelberg, Germany). A total volume of 0.5 mL blood was obtained using three to five kissing bugs—*Rhodnius prolixus* (donation from Dr. Sci. Claudio Lazzari, University of Tours, Tours, France) (K)— and the same volume was taken using medicinal leeches—*Hirudo medicinalis* (Biopharm, Hendy, South Wales, UK) (L). Two to three medium-sized unfed leeches were manually applied on individual animals depending on their body weight (two leeches in animals up to 10 kg and three for animals above 10 kg). The sampling time for kissing bugs and leeches was 10 to 15 min. A more detailed description of procedure implementation [14,40] and validation has been presented previously [15]. Saliva samples were collected with swabs (Salivette^®^, Sarstedt AG & Co, Nümbrecht, Germany), and tears were collected with Schirmer slips (Schirmer Tear Test Strips, Eickemeyer^®^, Tuttlingen, Germany). A tuft of hair was collected by cutting directly next to the skin in the shoulder area.

After samples were taken, the ibex was placed in a cage for five minutes to calm down. When no visible signs of stress (flattened ears, increased respiratory rate, dilated pupils) were observed, it was released into the group. Some of the ibexes urinated on the floor when they were in the cage. The urine was collected, and a fresh substrate was prepared for each ibex.

Faeces were collected during the same period (the day before to avoid increased C concentration as a result of capturing and collecting other samples). The ibexes were observed, and samples were collected immediately after defecating without agitating the animals.

The entire procedure (capturing and sampling) was coordinated by a qualified zoo animal veterinarian who also performed venipuncture.

All samples were labelled to ensure traceability and transported to the laboratory at 4 °C and stored at −20 °C until further analysis of C concentration. To obtain sera, blood was centrifuged at 721× *g* for 10 min before storage.

Samples from different matrices were collected from 2020 to 2022, from July to October. In 2020, we collected samples from twenty-nine ibexes. In the first year, we were unable to obtain tears or urine from any of the animals. In August 2021, urine sampling was performed in twelve ibexes, and urine was successfully obtained from six ibexes. In September 2022, tear sampling was performed in twelve ibexes, and tears were obtained from two ibexes. We could only compare the results from different matrices in the same animal in nineteen ibexes. Samples taken in July and August were combined into the summer group, and samples taken in September and October were combined into the autumn group.

Date of collecting the samples and number of samples collected from ibexes (Table 1).

The diagnostic procedure for annual monitoring was represented by Council Directive 92/65/EEC (BALAI) [41], valid up until April 2021, and thereafter by Regulation (EU) 2016/429 (“Animal Health Law”) [42]. For this reason, biochemical and haematological profiles from sera [15], parasitological examination from faeces, and some other laboratory procedures for important diseases that were also performed are not included in this study.

### 2.2. Laboratory Tests

Concentrations of C were measured in serum (three different procedures: V, K, L), saliva, hair, urine and tears, and faecal glucocorticoid metabolite (FGCM) concentration was measured in faeces, using the commercial enzyme immunoassay Cortisol ELISA (Demeditec Diagnostics GmbH, Kiel, Germany) according to the manufacturer’s instructions and validated as described previously [10,30,43,44].

### 2.3. Data Analysis

After data collection, the mean values of C concentration in different categories, sexes and ages of Alpine ibexes were calculated. Ibexes were grouped according to age into three groups: ten months and younger, between ten months and ten years, and older than ten years. Due to the small sample size and non-normal data distribution in some of the variable subgroups (Kolmogorov–Smirnov test), nonparametric tests (either Mann–Whitney test, *p* < 0.05 or Kruskal–Wallis test, *p* < 0.05) were used to find statistically significant differences between independent groups (i.e., sex, age, seasonal sampling). When we found significant differences using the Kruskal–Wallis test, we then applied a series of Mann–Whitney tests with Bonferroni correction. Additionally, effect sizes (r = Z/√n) were calculated.

In the cases where the samples were taken from the same animal (N = 19), a paired nonparametric test was used (Wilcoxon test, *p* < 0.05).

Statistical Package for the Social Sciences (SPSS), version 26, was used for all analyses.

Abbreviations in the text (Table 2).

## 3. Results

All 29 Alpine ibexes were healthy. No clinical signs were observed at the time of the study. No signs of stress or behavioural changes were observed the day after capture and sampling. Concentrations of C collected from different matrices were compared (Table 3).

The highest mean C (or FGCM, accordingly) concentration was measured in serum (V, K, L), followed by hair and faeces, and the lowest mean value was found in saliva.

Blood collection from a single animal using venipuncture (V) was compared with collection using kissing bugs—*Rhodnius prolixus* (K)—and medicinal leeches—*Hirudo medicinalis* (L) (Figure 1).

The mean C concentration in serum collected using venipuncture was the highest (95.10 ng/mL) (min = 62.08 ng/mL; max = 157.55 ng/mL), and the concentration in serum collected with kissing bugs was the lowest (51.35 ng/mL) (min = 17.22 ng/mL; max = 85.39 ng/mL). Statistically significant difference was found in the C concentration between venipuncture and kissing bugs (Wilcoxon: Z = −3.823, *p* < 0.001). The mean C concentration collected with leeches was 78.29 ng/mL (min = 31.93 ng/mL; max = 137.21 ng/mL). No statistically significant difference was found compared with venipuncture (Wilcoxon: Z = −1.650, *p* = 0.297). On the other hand, a statistically significant difference was found in the C concentration between kissing bugs and leeches (Wilcoxon: Z = −3.179, *p* = 0.004).

Concentrations of C in a single ibex in saliva, faeces, and hair were compared (Figure 2).

The mean C concentration was highest in hair (29.63 ng/g; min = 8.23 ng/g; max = 52.60 ng/g) and lowest in saliva (12.63 ng/mL; min = 2.42 ng/mL; max = 23.56 ng/mL). The mean value of FGCM concentration in faeces was 21.68 ng/g (min = 9.75 ng/g; max = 39.82 ng/g). Statistically significant differences were found between saliva and faeces samples (Wilcoxon: Z = −2.495, *p* = 0.034) and between saliva and hair samples (Wilcoxon: Z = −3.662, *p* < 0.001). The difference between faeces and hair was not statistically significant (Wilcoxon: Z = −1.972, *p* = 0.146).

Concentrations of C in urine and tears were measured in six and two ibexes, respectively. The mean C concentration in urine was 126.12 ng/mL (min = 85.75 ng/mL; max = 198.87 ng/mL), and the C concentrations in tears were 12.28 ng/mL and 14.28 ng/mL.

Various invasive and non-invasive methods of collection were used. The results are shown in Table 4, Table 5 and Table 6. The differences in C (or FGCMs, accordingly) concentration by sex, age, and sampling season (summer and autumn) are presented.

The concentrations of C (or FGCMs, accordingly) obtained from different matrices by sex are presented in Table 4.

A significant difference in C concentrations between males and females was found in saliva (*p =* 0.023). No significant differences between sexes in serum (V), faeces, or hair were detected (all *p* > 0.05).

The concentrations of C (or FGCMs, accordingly) collected from different matrices by the age of the ibexes (three groups) are shown in Table 5.

Significant differences (*p =* 0.002) in C concentration among the three groups of ibexes were measured in serum (V). No significant differences were found among the three groups in saliva, faeces, or hair. Mann–Whitney pairwise comparisons with Bonferroni-corrected *p* values regarding faecal data revealed a statistically significant difference between ibexes younger than 10 months and those older than 10 years (Mann–Whitney: Z = 21.89, *p* = 0.001).

The concentrations of C (or FGCMs, accordingly) collected from different matrices according to the season of sampling (summer/autumn) are presented in Table 6.

A highly significant difference was found between C concentrations in hair sampled in summer and autumn (*p* < 0.001). No significant differences were found between C concentrations in serum (V), saliva, or faeces sampled in summer compared to those sampled in autumn. Although the difference in serum (V) was not statistically significant (*p =* 0.073), the effect size was notable (r = 0.24).

## 4. Discussion

Animal health regulations establish specific requirements and preventive measures for disease control in captive ibex populations. Therefore, various diagnostic, disease monitoring, and animal welfare procedures are routinely implemented. Concentrations of C increase in stressful situations; for example, high concentrations can be found in animals in confined environments, those that present subclinically or clinically apparent infection or parasitic and metabolic diseases. Health protocols [29,37,45] and animal welfare [15] are an important part of the strategy at our zoo. In addition to welfare, we pay great attention to the non-invasive sampling of animals, their appropriate nutrition, and knowledge regarding individual and group behaviour.

Concentrations of C are routinely determined from blood, hair, saliva, faeces, and milk [17,19,20,23,24]. The concentration of C can be measured in blood as well as in other matrices that cause less stress and are often safer for both the animals and those collecting the samples [9,12,15,20].

In our study, C concentration was measured in ibexes in eight different matrices: serum (V, B, and L), saliva, faeces, hair, urine, and tears. We had the greatest difficulty collecting blood using kissing bugs and collecting urine and tear samples.

### 4.1. Blood Collection

There are many studies on the concentration of C in serum or plasma in domestic animals [46,47,48], but to the best of our knowledge, no one has studied C concentrations in the Alpine ibex. Studies have been performed in the Alpine ibex for detecting C in hair [18] and FGCMs in faeces [49,50]. Sartorelli et al. [51] measured C concentrations in the blood cortisol of ibexes. The results were not comparable with ours because a different assay was used. The serum C concentration in their study was measured using RIA (Sorin Biomedica), whereas we used the commercial enzyme immunoassay Cortisol ELISA (Demeditec Diagnostics GmbH, Kiel, Germany). The use of rapid tests to measure C concentrations was a disadvantage, in that we could usually only compare the relevant results within one group.

Due to the stressful nature of capture, serum collected using venipuncture is a suitable medium for determining C concentration when blood is to be collected for other purposes, such as antibody monitoring or haematological and biochemical analyses as part of diagnostic procedures. Alternatively, other techniques, such as blood collection with medical leeches [15] and kissing bugs [13], can be used.

In our study, we found significant difference (*p* = 0.002) in serum (V) C concentrations between younger, adult, and older ibexes. Mean values increased with age, indicating that the older animals were the most stressed. Several factors could be responsible for this result, such as the sexual activeness of the group, previous capture experience, or the rank order of the animals. Further study would be needed to determine the reasons for the difference between groups.

Blood collection with medicinal leeches is a good alternative to other more complex and invasive methods, and could represent a significant advance in blood collection for preventive medicine and epidemiological studies in zoo animals [29], e.g., for the re-emergence of the bluetongue virus, which is a very important pathogen [52]. Medicinal leeches can indeed be used as a reliable blood collection tool for haematological and biochemical health assessment [15]. In our study, no statistically significant differences were found between serum C concentrations collected with leeches (M = 77.04 ng/mL) and venipuncture (M = 93.49 ng/mL). Our results, as well as other data [15], confirm that leeches can be a useful and less invasive method for blood collection from ibexes and other zoo animals.

Although methods of successful blood collection with kissing bugs (Triatominae species) have been described [13,14,40], we found statistically significant differences between venipuncture and kissing bugs and between leeches and kissing bugs. The mean value of serum C was higher in serum collected using venipuncture and leeches and much lower in serum collected with kissing bugs (M = 52.86 ng/mL). The lower C concentration in kissing bugs may be explained by poor blood collection technique. Blood collection from kissing bugs is mandatorily performed with a needle [14]. If this process is performed carelessly, the contents of the gastrointestinal tract may enter the test tube, significantly affecting the concentration and lowering the actual concentrations, and this could be one of the reasons for which our results are not comparable to those obtained with venipuncture and leeches. The second reason could be the small amount of serum obtained. Blood was taken with a total volume of 0.5 mL. However, Markvardsen et al. mentioned that another species of kissing bug, *Dipetalogaster maximus*, is able to suck up to 4 mL of blood as an adult. On the other hand, it has been confirmed that a sample of at least 500 µL of full blood is sufficient for clinical chemistry, and at least 300 µL of plasma is required for haematology [40]. Kruszewicz et al. [14] reported that the maximum quantity blood that can be drawn at one time from the front of an adult individual or an individual in the last larval stage ranges from 1.1 to 3.8 mL. Both genders of kissing bug and all five developmental stages feed on blood. Usually, males of nymphs 4 or 5 (L4-L5) are used. Kissing bugs prefer dry, shaded gaps that are as tight as possible, which allow them to calmly wait out the day to feed under cover of night. After kissing, blood is stored in the anterior part of the intestine centre, before sliding back into the stomach. The watery excretion of the blood fraction concentrated in the digestive tract begins at the end of the bug’s feeding on the host, i.e., after about 20 min [14]. Therefore, quick capture of the insect immediately after the collection of blood from the patient and the technique of collecting and handling kissing bugs to obtain blood is very important. Kissing bugs were less responsive at lower ambient temperatures, which is also a reason why we did not obtain enough blood in some cases, especially during autumn. All these facts could be reasons that, in our study, we obtained statistically significant results that were comparable with the results of collecting blood using venipuncture and with leeches.

Blood collection with medical leeches or kissing bugs could be performed without needing to capture the animals. The wooden platform can be placed at the location at which the target animal usually rests. Leeches or kissing bugs can be placed in a closed container with a net or membrane (Parafilm M, Bemis, Shirley, NY, USA) on top and attached below the mattress. After the ibexes lie down on the platform, leeches or kissing bugs will bite and start feeding. With this method, the ibexes do not know they have been exposed to leeches or kissing bugs, and when they leave the platform, the leeches or kissing bugs, full of blood, can be removed. Unfortunately, we did not include such a procedure in our study. Closing an animal in a box with leeches placed next to them also works in some other species, and is safer and less stressful than venipuncture under manual restraint or anaesthesia [15,29].

### 4.2. Saliva, Faeces, and Hair Collection

Some important rules must be kept in mind when interpreting the results of C concentration in different matrices. The concentration of C in most matrices other than hair may also depend on a circadian rhythm, diurnal rhythm, and season. Aurich et al. [53] studied the effects of season, age, sex, and housing on salivary C concentrations in horses. Concentrations of C follow a diurnal rhythm, and increase during the active breeding of animals. Therefore, the time of day and the reproductive status of the horse are important in experiments involving the analysis of C in saliva. All animals in our study were sampled in the morning to exclude the influence of circadian rhythm. It must also be mentioned that a habituation effect of C concentrations between seasons could influence.

Habituation to C was conceptualised as a change in total C output between the first and second sampling, with a greater decrease in the index indicating greater degree of habituation. This is a reason to be attentive when interpreting the results and comparing them with both one’s own results and those of other researchers.

Shutt et al. examined the effects of research activities on levels of FGCMs, a proxy for indicator of physiological stress, in wild western lowland gorillas. They examined how human proximity affects FGCMs in recently and long-term habituated groups. The group habituated to humans had the highest FGCMs, suggesting a cumulative FGCM response consistent with descriptions of a hormonal adaptation response to a chronic intermittent stressor [54]. These findings may provide important information for interpretation of results, especially in zoo animals that are exposed to constant contact with humans.

Palme [55] found that faecal sampling and FGCM analyses may seem simple and straightforward, but researchers must select and apply these methods properly. They must also be aware of the many pitfalls and potentially confounding factors, and last but not least, they must interpret the results carefully. For this reason, we avoided comparing results obtained with different methods or in different species in our study. Touma and Palme [19], who developed a non-invasive technique for the detection of FGCMs in faecal samples, offered a solution in order to overcome the drawbacks of the technique. However, quantitative comparison of the results should be avoided, because in commercial tests, it is usually immunoreactive substances that are being measured, which most probably differ between the matrices (cortisol vs. cortisol metabolites). The faecal samples also represent the situation without the need for stressful manipulation. The values of other matrices could increase due to the capture of animals during sampling. In our study, faecal samples were collected the day before to avoid increased C concentration as a result of capturing and collecting other samples.

Saliva and faeces can be used to measure C at a single point or within 12 h (acute stress), whereas performing C analysis on hair reflects secretion over longer periods of time and may be associated with chronic stress [18,56]. In our study, samples were always collected in the morning, and the capture procedure was the same. Ibexes were captured for all collected samples except for the collection of faecal samples.

It is known that C concentration depends on the animal species. The FGKM concentrations found in wild animals and related domestic animal species were published recently [33]. Since different studies have used different assays and different methods of dilution and extraction, as well as differing in terms of whether they use dried or fresh faeces, the possibility of direct comparison is limited.

Faecal samples are useful materials for sampling [18] because we can avoid the stress caused to animals caused by capture. Brivio et al. [28] did not measure any significant change in the concentration of male FGCMs during chemical immobilization, and they confirmed that immobilization did not cause serious changes in C concentration that could potentially be dangerous to individual well-being. Dulude-de Broin et al. [57] found that sex had no detectable effect on FGCMs, but hair C concentrations were lower in adult males than in adult females. In the group older than ten years, we found in serum (V), statistically significantly lower C concentrations in males than in females (*p* = 0.002; N = 13), but the results in saliva, faeces and hair were not significantly different. Apparently, there was an unknown stress factor occurring in our older group. According to the results, the reason for the increased C concentration could be animal manipulation, because we obtained increased values only in saliva and serum (V), and not in hair, where chronic stress would be detected, or in faeces, where there was no animal manipulation. Obviously, older animals are more exposed to stress during capture.

We measured a significant difference in salivary C concentration between females and males in the whole group. In addition to higher C concentrations in saliva, salivary secretion was also found to be higher in males than in females. In the whole group, no significant differences were found in the C concentration in hair or FGCM concentration, which is probably because we detected the secretion of C in saliva at the time of sampling, whereas hair and faecal samples showed a delayed response to stress that it is not yet possible to detect.

Our research results show that when interpreting the C concentration in hair, the method and location of hair removal are also important [31,57].

It is necessary to distinguish between different types of hair. The fur hairs of most animals stop growing once they reach a reasonable length, as the hair follicle enters a resting state called the telogen phase. An additional complicating factor for hair analysis in nonhuman animals is cyclic hair growth. In most animals, hair growth stops after a certain time, and hair falls out twice a year. Hair stores C for a longer period than originally thought, although the resulting concentrations are probably not useful in a context where C is considered a stress-sensitive biomarker. There is ample evidence for glucocorticoids, but little-to-no evidence that hair contains a record of past/historical stress [31]. In our study, we always cut right next to the skin in the shoulder area, so the results were comparable within the study.

Dulude-de Broin et al. [57] found that age had no detectable effect on FGCMs, but hair C concentrations were higher in younger than in older goats. In our study, no significant differences were found among ibexes younger than ten months, adults, and ibexes older than ten years in saliva, faeces and hair. We found a statistically significant difference only in faecal samples between ibexes younger than ten months and older than ten years. Our results indicate acute stress in older ibexes.

Prandi et al. [18] reported that the concentration of C in hair in Alpine ibexes is 22.40 ng/g. They found a significant difference between sexes. In our study, a highly significant difference in hair (*p* =< 0.001) was measured between samples collected in summer and autumn, with higher concentrations being observed in autumn. The mean hair C concentration (35.95 ng/g) could indicate chronic stress, but nothing out of the ordinary happened in the group during this period.

In addition to differences in the concentration of C in individual species, differences in sampling, such as the type of medium (e.g., hair, faeces), the quality of the sample (e.g., hair/undercoat, dry/fresh faeces), and the choice of enzyme immunoassay, also affect the results. Jewgenow et al. [21] described differences in concentrations observed when using different enzyme immunoassays. They found differences in the mean value for hair C levels for some mammalian species determined using the cortisol-21-HS and cortisol-3-CMO enzyme immunoassays. The Alpine ibex was not included in this study. This fact is important for the comparison and interpretation of the results of different animal species and different matrices of the same species.

In our study, the mean FGCM concentration for the ibexes was 22.59 ng/g, while in hair it was 27.40 ng/g, and in saliva it was 12.78 ng/g.

Cortisol metabolites have been identified in many cervid and bovid species [55], including the Alpine ibex [28], so we can safely conclude that metabolites were also present in our samples. However, we did not perform biological or physiological validations because the ibex is a wild, rare, and endangered species. In addition, animal testing is generally problematic under current legislation; there is not the slightest chance of obtaining a permit for such experiments. Capturing an ibex in the Julian Mountains and collecting their faeces is not possible for us, although researchers in Switzerland have managed to collect ibex samples in the past [28]. The same problem exists with zoo animals. Capturing and collecting samples is very stressful for ibexes. Since our method has not been validated for this species (ibexes), in our article, we compared the results of individual animals within the group in terms of age differences, the differences between sexes, and the season. We compared C concentrations in the same animal but sampled it in different ways (that is, with different matrices).

### 4.3. Tear and Urine Collection

There are no studies on C in the urine or tears of Alpine ibexes. The collection of urine and tear samples is a laborious procedure, and is therefore less acceptable or even unacceptable for the purposes of C determination or other diagnostic procedures.

Urine collection by catheterisation was unsuccessful without sedatives in ibexes. Urine was collected from the cage floor after other samples were taken. This method is non-invasive, but it depends on the physiological need for urination. This is the reason that samples were taken from only half of the animals examined (6/12) in our study. Tear collection with capillary tubes (0/29) or with Schirmer slips (2/12) was also not successful. When using these two methods, samples of insufficient volume (about 50 µL) were obtained.

The mean C concentration in urine (No. = 6) was 126.12 ng/mL, which is even higher than the mean C concentration in blood. One explanation for this difference is that C tends to be found more extensively in urine, in the so-called “urine free cortisol”, than in serum (V) or saliva [58].

The concentration of C in tears was measured in two ibexes. The concentrations of C in tears were 12.28 ng/mL and 14.28 ng/mL, values which are slightly higher than those found in saliva (mean value = 12.78 ng/mL). This most likely means that C secretion in tears is similar to that in saliva.

These two methods are unsuitable for sampling ibexes.

## 5. Conclusions

Catching and invasive sampling methods in ibexes involve a lot of stress, not only for the individual animals, but also for the group as a whole. Non-invasive blood sampling methods have the potential to substantially improve welfare in sampled animals. Non-invasive blood sampling with leeches is a good alternative when venipuncture is impractical or would be too stressful for the animal. A statistically significant difference in serum C concentration was found between venipuncture collection and using kissing bugs. Therefore, the technique of blood collection with kissing bugs and their proper handling is very important. Urine and tear sampling are laborious procedures, and are therefore less suitable for C determination. A highly significant difference in C concentration between samples collected in summer and autumn was found in hair. A significant difference in C concentration between male and female animals was found in saliva. A significant difference in C concentration was found in serum samples (V) between ibexes younger than ten months and older than ten years. In addition, finally, a significant difference in FGCM concentration was found in faecal samples between ibexes belonging to the groups younger than ten months and older than ten years.

## Figures and Tables

**Figure 1 animals-13-02491-f001:**
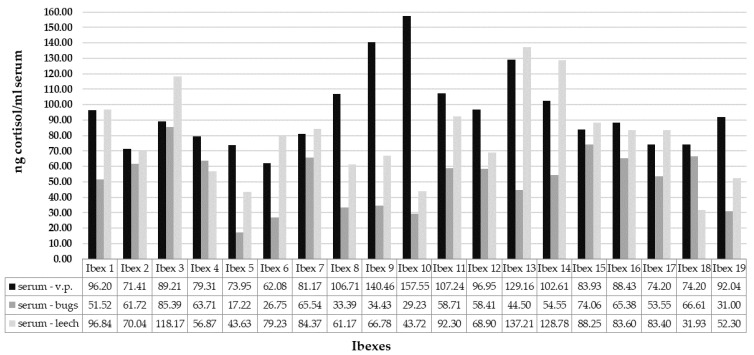
Differences in serum C concentrations collected with venipuncture (V), kissing bugs (K) and medicinal leeches (L).

**Figure 2 animals-13-02491-f002:**
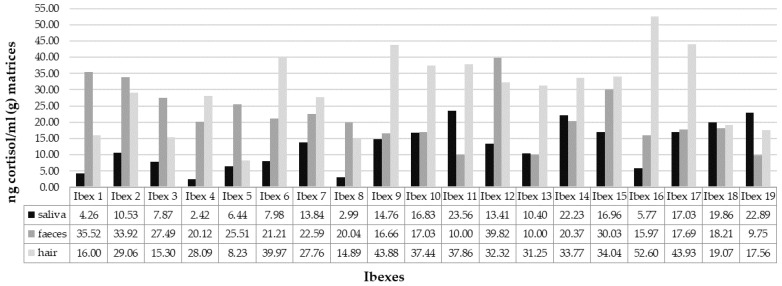
Differences in C (or FGCMs, accordingly) concentrations in saliva, faeces, and hair.

**Table 1 animals-13-02491-t001:** Collection of samples from July 2020 to October 2022.

Year of Sampling	2020	2020	2021	2022
	July August	September October	July August	July August
Serum (V)	10	22	12	12
Serum (K)	8	10	1	10
Serum (L)	1	10	4	8
Saliva	/	18	11	/
Hair	10	21	12	/
Faeces	19	19	5	/
Urine	/	/	6	/
Tears	/	/	/	2

**Table 2 animals-13-02491-t002:** Explanation of abbreviations.

	Abbreviation
Minimum value	min
Maximum value	max
Mean value	M
Standard error	SE
Standard deviation	SD
Kruskal–Wallis test	Hi^2^
Standardised score of Z-test	Z
Statistical significance	p
Degrees of freedom	df
Effect size	r
Number of tested samples	N

**Table 3 animals-13-02491-t003:** Differences in C (or FGCM, accordingly) concentrations in different matrices.

Total	Serum—V (ng/mL)	Serum—K (ng/mL)	Serum—L (ng/mL)	Saliva (ng/mL)	Faeces (ng/g)	Hair (ng/g)
Mean	93.49	52.86	77.04	12.63	22.59	27.40
SE	3.45	4.29	5.50	1.14	1.53	1.92
SD	25.85	23.08	26.37	6.16	10.04	12.59
N	56	29	23	29	43	43

**Table 4 animals-13-02491-t004:** Differences in C (or FGCMs, accordingly) concentrations from different matrices according to sex.

Matrices	Sex						Mann–Whitney		Effect Size
	Males			Females					
	M	SD	N	M	SD	N	*Z*	*p*	r
Serum (V)	89.71	24.03	25	96.53	27.23	31	−1.39	0.164	−0.19
Saliva	10.12	6.31	15	15.32	4.86	14	−2.27	0.023	−0.42
Faeces	23.66	9.88	22	21.47	10.33	21	−0.91	0.362	−0.14
Hair	25.98	11.05	20	28.64	13.92	23	−0.44	0.661	−0.07

**Table 5 animals-13-02491-t005:** Differences in C (or FGCMs, accordingly) concentrations from different matrices according to age.

Matrices	Age									Kruskal–Wallis Test		
	Ten Months and Younger			Ten Months to Ten Years			Older than Ten Years					
	M	SD	N	M	SD	N	M	SD	N	Hi^2^	df	*p*
Serum (V)	78.08	19.23	16	94.26	24.32	27	110.85	25.96	13	12.94	2	0.002
Saliva	15.14	6.01	10	11.71	5.17	14	10.20	8.43	5	2.96	2	0.228
Faeces	17.56	7.96	11	23.09	9.30	20	26.38	11.69	12	4.88	2	0.087
Hair	31.05	15.18	13	25.64	11.95	19	26.14	10.37	11	1.21	2	0.546

**Table 6 animals-13-02491-t006:** Differences in C (or FGCMs, accordingly) concentrations in different matrices according to the season of collection (summer/autumn).

Matrices	Period						Mann–Whitney		Effect Size
	Summer			Autumn					
	M	SD	N	M	SD	N	Z	*p*	r
Serum (V)	88.43	19.55	34	101.30	32.30	22	−1.80	0.073	−0.24
Saliva	12.61	6.82	11	12.64	5.92	18	0.02	0.982	<0.01
Faeces	23.18	9.53	24	21.86	10.88	19	−0.43	0.668	−0.07
Hair	19.24	7.90	22	35.95	10.83	21	−4.52	<0.001	−0.69

## Data Availability

Data is contained within the article.

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
