# Peer review of "Comparison of Cortisol Concentrations in Different Matrices in Alpine Ibex (Capra ibex) at the Zoo"

_animals, 2023, doi:10.3390/ani13152491_

Round 1

Reviewer 1 Report (Previous Reviewer 1)

Thank you for modifying your manuscript as suggested, it looks much better than before. I would add just few comments:

line 271: Captivity, with potentially high concentrations of animals, causes direct contact and stress, which can affect the incidence, transmission, and severity of disease. I would change with: High concentrations of animals with a reduced environment could potentially cause stress, influencing the incidence, the transmission by direct contact, and the severity of disease.

line 324: Blood collection with a needle is mandatory. I would change with: Blood collection with a needle from kissing bugs is mandatory.

line 351-361: (...) We did not find sex differences in other media. I suggest to explain briefly why.

line 374: Concentrations of exogenous CORT in hair extracts from captive mountain goats (Oreamnos americanusfrom were studied. Please remove “from”.

Thanks to the proofread, the manuscript looks clearer than before.

Author Response

Dear REVIEWER

Thank you again for your comments.

Kind regards,
author and co-authors

Reviewer 2 Report (Previous Reviewer 2)

Given that this is the second submission of this manuscript for review, the authors should be commended on addressing many of the previous issues that were raised with the paper. The manuscript reads markedly better and the science is improved when compared to the previous submission. The content of the paper also seem more relevant to the field as a whole with the streamlining of the writing that was done.

However, there are still some issues that remain which the authors must address. The discussion and conclusions are the most problematic aspects of the paper. The authors should make their findings more relevant to the reader by making explicit connections between the literature and their results rather than presenting the results and expecting the reader to interpret them on their own. The specifics of this are outlined in detail in the comments provided. There are also a few minor errors and clarifications that should be addressed but on the whole, apart from the discussion and conclusions, the manuscript is a marked improvement on the preceding version.

It is for these reasons that I am recommending that the manuscript be reconsidered following major revision.

Some of the language use could be improved.

Author Response

Dear REVIEWER,

thank you again for all your valuable comments. We appreciate that you took the time and made the effort to review our manuscript. You have raised important issues. In atachment are our responses to each comment. We hope that we have responded satisfactorily. The article was proofread by MDPI servis.

With regards

Reviewer 3 Report (New Reviewer)

Comparison of cortisol concentrations in different media in Alpine ibex (capra ibex) at the zoo

Keeping animals in enclosures may be stressful for these individuals and monitoring various parameters of stress (including stress hormones such as cortisol or cortisol metabolites) is important regarding animal welfare aspects.

General remarks concerning the manuscript:

The study describes the measurement of cortisol concentrations in various types of samples (blood, saliva, faeces, hair, urine, and tears) and compared them. In blood, the cortisol concentrations were compared in samples collected with three methods (venipuncture, kissing bugs and medicinal leeches). However, no clear aim is shown, why this experiment was performed.

Another point is a methodological aspect. For measuring cortisol and cortisol metabolites, the most often used assay procedures are immunoassay or HPLC in combination with mass spectroscopy (MS). As the costs of immunoassays are lower compared to HPLC/MS, this method is mostly used. Most of the (commercially) available immunoassays are tested for measuring cortisol in (human) plasma or serum samples and not validated for use in domesticated, zoo and wildlife animals or in matrices as for example urine or faeces. Therefore, as a prerequisite for measuring stress hormones or their metabolites, a validation of the assay(s) for use in the respective species and matrices has to be done. In case of measuring steroid hormone metabolites using immunoassays it is advisable to select an assay with high cross-reactions with the dominating metabolites. This is particularly important, as in faeces and urine predominantly cortisol metabolites are excreted. In some animal species the metabolism of cortisol is so intensive that for example in sheep only trace amounts of (authentic) cortisol are present in the faeces (Möstl, E., Maggs, JL., Schrötter, G., Besenfelder, U., Palme, R. (2002): Measurement of cortisol metabolites in faeces of ruminants. Vet. Res. Commun. 26, 127-139.). The “cortisol” measured by immunoassay (esp. in urine or faeces) may not be cortisol, but other immunoreactive substances considered as “immunoreactive” cortisol. A validation of the test system will demonstrate that the values measured are not only “blank values”, which may be the case measuring complex matrices like faecal samples.  For evaluation, if the assay used is measuring cortisol or cortisol metabolites, an “immunogram” can be made.  A second  test for validating assays for glucocorticoids or glucocorticoid metabolites is a biological validation. After a stressful situation samples are collected in an appropriate time and the increase of substances is monitored. An alternative biological test procedure is to inject a synthetic glucocorticoid (not cross-reacting in the assay used) and to monitor, if the immunoreactive substances measured decrease.

Therefore, if cross-reacting substances are present in a substantial amount, immunoassays which differ in the percentage of cross-reaction with this metabolite will show big differences in concentrations measured. Therefore, the results of an immunoassay without testing the validity of the test system in a given matrix should be considered only as “immunoreactive” cortisol (independent whether the immunoreactive substance originates from cortisol or from another source).

In the manuscript, no data concerning the validity of the assay for Alpine ibex and on matrices like faeces and urine are presented. The publications cited as validation concern pigeons, horses (blood samples) and cattle (hair samples). However, for the study presented, there is no proof whether the assay used reacts with cortisol or cortisol metabolites, and validation is missing.

Quantitative comparison of the results of this study with the test used should be avoided, as 1) only immunoreactive substances were measured which most probably differ between the matrices (cortisol vs cortisol metabolites) and 2) the time factor has to be considered. As the sampling procedure including capturing and translocation most probably was stressful, I assume, that saliva and blood samples (including samples from kissing bugs leeches) are from “stressed” animals. The faecal samples, however, represent the cortisol situation about 12 h earlier (not sure about ibex, but in sheep the gut passage time is about 12 h) and therefore represent the situation before the stressful manipulation.

Cortisol in hair is considered as a window into the past in humans, where more than 80 % of the hair are in the anagen phase, whereas in most animals the hair growth stops after a certain time and shedding occurs twice a year. The hypothesis, that cortisol is stored in the growing hair and remains there for a longer period of time is now in question (Hair glucocorticoids are not a historical marker of stress – Exploring the time-scale of corticosterone incorporation into hairs in a rat model General and Comparative Endocrinology 341(1):114 Pernille Colding-Jørgensen et al., 2023).

Additional points:

“media” should be replaced by “matrices”.

Lines 66-68: I recommend re-wording, as sampling procedures do not contribute to the welfare of animals.

Lines 117-120: I do not agree that an influence of a stressful situation and consequently the secretion of cortisol into the blood were avoided, as these animals were already stressed by capturing and translocation.

In the section “discussion” it should be mentioned that a habituation effect on cortisol concentrations may occur and differences between seasons may be influenced by habituation.

Concerning the abbreviation CORT for cortisol please see: Haff (2016).

Finally, a statement of the ethic committee is needed, but missing.

no comments

Author Response

Dear REVIEWER,

thank you for all your valuable comments. We appreciate that you took the time and made the effort to review our manuscript. You have raised important issues. In atachment are our responses to each comment. We hope that we have responded satisfactorily. The article was proofread by MDPI servis.

Round 2

Reviewer 2 Report (Previous Reviewer 2)

The authors have addressed the comments raised. I am therefore recommending that the manuscript be accepted for publication.

Author Response

Dear REVIEWER

Thank you again for your comments.

Kind regards,
author and co-authors

Reviewer 3 Report (New Reviewer)

Comparison of cortisol concentrations in different media in Alpine ibex (capra ibex) at the zoo

The new version shows an improvement of the manuscript. However, my main objection was not addressed: You used an assay without validation in ibex. For validation, you cite studies analysing samples of pigeons, horses and cows, but no paper using this assay in ibex. Please see the paper of Touma and Palme (2005): Measuring fecal glucocorticoid metabolites in mammals and birds: the importance of validation. Ann N Y Acad Sci. 2005; 1046:54-74.

For faecal samples, you cannot be sure that the assay pics up any glucocorticoid metabolite. Therefore, it is not justifiable to recommend the assay for measuring glucocorticoids or glucocorticoid metabolites in faeces.

Minor remarks:

Line 30: please add “immunoreactive” cortisol (iC). This will inform the reader, that the intention of the paper is not the measurement of authentic cortisol but cortisol plus immunoreactive cortisol.

Line 105: please consider the new literature: Cortisol in hair is considered as a window into the past in humans, where more than 80 % of the hair are in the anagen phase, whereas in most animals the hair growth stops after a certain time and shedding occurs twice a year. The hypothesis, that cortisol is stored in the growing hair and remains there for a longer period of time is now in question (Hair glucocorticoids are not a historical marker of stress – Exploring the time-scale of corticosterone incorporation into hairs in a rat model General and Comparative Endocrinology 341(1):114 Pernille Colding-Jørgensen et al., 2023).

Line 130: You may inform the reader that in blood cortisol is mainly bound to binding proteins (cortisol binding protein and albumin), only a minor fraction is non-bound cortisol (free cortisol). The protein-bound cortisol is too big and too polar to pass the cell membranes and cannot reach the intracellular receptors. Free cortisol can pass the membranes and cortisol in saliva is a parameter for the free cortisol fraction (receptor active fraction) in blood.

Line 188: please show g-value instead of rpm

Line 349: Concerning measuring Cortisol in blood of ibex, there is a paper available: Alberto Prandi, Tanja Peric, Mirco Corazzin , Antonella Comin, Monica Colitti: A first survey on hair cortisol of an Alpine ibex (Capra ibex ibex) population. Animal Science Papers and Reports vol. 36 (2018), no. 1, 57-74.

Author Response

Dear reviewer
Thank you again for
your helpful comments.

Kind regards author and co-authors

This manuscript is a resubmission of an earlier submission. The following is a list of the peer review reports and author responses from that submission.

Round 1

Reviewer 1 Report

Thank you for submitting your manuscript.
I think this study is innovative, interesting, and well written, even if the Material &Method part must be improved. So, I suggest some changes.
Below my considerations:
line 81-82: in the Introduction (or also in the Discussion may be...) it would be interesting to mention that other strategy to collect blood without stress exist, such as training. Zoo animals could be trained to these medical procedures even this could not be applied to wild animals, anyway I think the use of leeches in wild animals is not so easy to apply too....
line 105-115: I consider that part must be improved : Which kind of syringe do you used for venipuncture? During the capture procedure of each animal when did you put the bugs and the leeches on animal? How long was the procedure?

Reviewer 2 Report

The study described in this manuscript focuses on various methods of biological sample collection for the assessment of stress and reproductive hormones in Alpine ibex. The study has many issues which make interpretation of the study difficult and its relevance questionable. The authors have a potentially valuable dataset but the manner in which the study has been carried out or is described in the manuscript makes it difficult to see the value of such a manuscript in adding to the existing literature.

Firstly, the study clearly suffers from the authors not being English first-language speakers. This is evident from the manner in which the authors describe and explain aspects of both the theory and the practicalities of the study design. It undoubtedly complicates the interpretation of the study design and methodology for the reader. As such the authors should seek the assistance of English editing before submission of the manuscript.

With the above in mind, there is a certain amount of the manuscript’s contents which are confusing to the reader and which seem to be related more to the scientific aspects of the study than to the language used. The first major issue is that the study aim and the methodology do not align. The study aim is described as being comparison of various methods of biological sample collection for estimating hormone levels in captive ibex. However, the statistical analyses focus on contrasting hormone levels between age and sex classes within the sample and between two seasons (summer and autumn). Despite the statistical methods used, and the absence of relevant statistical findings, the authors then proceed to compare and contrast the different sampling methods in the discussion. There is also the inclusion of reproductive hormones in the analyses, an aspect that is lacking any clear relevance or motivation in the study design or the theory presented.

There is also a problem of sample size. The authors have two sampling techniques for which they obtained 2 and 6 samples respectively, both of which are too small to be included in the statistical analyses. To their credit, the authors do not attempt to include these in the analyses but their inclusion in the study findings seems dubious. Certainly, their inclusion as points of discussion around the merits of their respective sampling methods seems unfounded and should rather be limited to a sentence or two at most.

An associated problem is the structure of the sample used in this study as well. It is not clear how many samples were obtained from each subject and over what time-frame. The authors describe the sampling structure in an ambiguous manner (as they do for most of their methodology) which leaves the reader unsure of how much confidence can be placed in their findings and whether or not the methods are appropriate or being applied appropriately.

Finally, the study does not appear particularly novel and does not seem to add meaningfully to the existing literature on animal welfare, animal endocrinology or methodology for sampling animal hormones, in part because the study design is ambiguous but also because the authors offer little synthesis of their findings in relation to the existing literature in the field. While I am sensitive to the fact that these methods may not yet have been applied to Alpine ibex specifically, this alone cannot form the justification for the publication of such a study. It is therefore my recommendation that this study be rejected for publication in Animals.

The authors would do well to seek the assistance of an English editing service when preparing such manuscripts for submission to a peer-reviewed journal. While the English writing in the manuscript is not bad, there are critical aspects of the language use and vocabulary which cloud interpretation of the writing and may be misleading to the reader.